# Maintaining Engagement in Adults with Neurofibromatosis Type 1 to Use the iCanCope Mobile Application (iCanCope-NF)

**DOI:** 10.3390/cancers15123213

**Published:** 2023-06-16

**Authors:** Frank D. Buono, Kaitlyn Larkin, Quynh Pham, Diane De Sousa, William T. Zempsky, Chitra Lalloo, Jennifer N. Stinson

**Affiliations:** 1Department of Psychiatry, Yale School of Medicine, New Haven, CT 06510, USA; kaitlynalarkin@outlook.com; 2Centre for Digital Therapeutics, University Health Network, Toronto, ON M5G 2C4, Canada; q.pham@uhn.ca (Q.P.); diane.desousa@uhn.ca (D.D.S.); 3Institute of Health Policy, Management and Evaluation, University of Toronto, Toronto, ON M5T 3M6, Canada; chitra.lalloo@sickkids.ca; 4Department of Pain and Palliative Medicine, Connecticut Children’s Medical Center, Hartford, CT 06106, USA; wzempsk@connecticutchildrens.org; 5Department of Pediatrics and Nursing, University of Connecticut School of Medicine, Stores, CT 06032, USA; 6The Research Institute, The Hospital of Sick Children, Toronto, ON M5G 1X8, Canada; jennifer.stinson@sickkids.ca; 7Lawrence S. Bloomberg Faculty of Nursing, University of Toronto, Toronto, ON M5T 1P8, Canada

**Keywords:** Neurofibromatosis Type 1, contingency management, chronic pain, mobile application, iCanCope

## Abstract

**Simple Summary:**

Mobile applications are an effective method to promote self-monitoring of various diseases and disorders. However, maintaining active engagement and adherence in mobile applications can be difficult. We found that using an incentive based system increased an individual’s usage of the mobile application in multiple areas (e.g., number of articles read, number of goals set). In keeping individuals engaged in the mobile application they were more apt to review and use the content to learn how to self-monitor their pain symptoms and reduce their chronic pain. We suggest implementing incentive based systems to increase engagement and adherence for mobile applications.

**Abstract:**

Introduction: Neurofibromatosis Type 1 (NF1) is an autosomal dominant genetic condition in which chronic pain is a predominant issue. Given the rarity of the disease, there are limited psychosocial treatments for individuals with NF1 suffering with chronic pain. Using mobile applications can facilitate psychosocial treatments; however, there are consistent issues with engagement. Utilizing a mixed methodology, the current study evaluated the customized iCanCope mobile application for NF1 on increasing engagement through the usage of contingency management. Methods: A mixed methods study from a subset of data coming from a randomized clinical trial that occurred from January 2021 to August 2022 was undertaken. Two groups (iCC and iCC + CM) were exposed to the customized iCanCope mobile application in which engagement data were captured in real-time with daily check-ins for interference, sleep, mood, physical activity, energy levels, goal setting, and accessing article content (coping strategies). Additionally, semi-structured interviews were conducted to gain insight into the participants’ experience at the end of the trial. Results: Adults (N = 72) were recruited via NF patient advocacy groups. Significant differences were noted between the groups in total articles read (*p* = 0.002), goals achieved (*p* = 0.017), and goals created (*p* = 008). Additionally, there were significant differences observed between user-generated goals and those that were app recommended (*p* < 0.001). Both groups qualitatively reported positive feedback on the customized mobile application, indicating that continued usage and engagement of the mobile application were acceptable. Conclusions: Employing customized mobile applications for adults with NF1 along with contingency management can leverage self-managed pain treatments while providing auxiliary resources to this population.

## 1. Introduction

Neurofibromatosis Type 1 (NF1) is an autosomal dominant tumor suppressor syndrome that predisposes affected individuals to benign and malignant tumors which can increase their likelihood of chronic pain (CP) [1,2]. Recent research has indicated that up to 60% of adults with NF1 can have CP [2,3,4,5]. Pain can be associated with tumors; however, it is often not localized to a specific structural lesion and can affect multiple regions of the body (e.g., migraines [6,7], lower back pain [8]. Medical treatment of individuals with NF1 is challenging due to the complexity of this disease [4,9]. Pharmacotherapy and/or surgery are often the first-line treatment for chronic pain [10,11,12,13,14,15,16,17] with up to 55% of adults with NF1 having been prescribed opioid medication during their lifetime [18].

Treatment for CP is best understood within a biopsychosocial framework which requires an interdisciplinary and multidimensional approach that incorporates pharmacological, physical, and psychological (mind-body) interventions [19,20,21]. Employing psychosocial or psychological treatments (e.g., Cognitive Behavioral Therapy (CBT)) is widely accepted as the most empirical approach to addressing the complexity of CP [22,23,24]. CBT guides individuals through psychoeducation about pain, cognitive restructuring (coping strategies), problem resolution, goal setting, and behavioral activation to enhance patients’ sense of control of pain, cultivate self-efficacy, eliminate the original negative coping model, strengthen skills training, and promote relaxation [25,26,27]. Within NF1, the existing research related to these treatments is suboptimal due to a variety of factors including few rigorous large-scale clinical trials, high attrition rates or drop-out rates, poor access to care, and the severity of the disease.

Mobile technologies can be applied to enhance the accessibility of CBT [28,29]. These technologies can empower individuals to take an active role in managing their condition by providing “in-the-moment” access to pain management techniques. In addition, users are able to personalize and tailor the content to meet their special needs. Additional benefits of mobile applications can include: improving scalability, cost-effective delivery, providing real-time convenience, and alternatives to the barriers required to access trained professionals [30,31,32]. Given the high preponderance of individuals having access to smartphone use, digital interventions have shown initial promise as a precursor, adjunct, and/or replacement for in-person treatment [32]. However, the psychosocial treatment methods employed in apps tend not to include empirically validated treatment modalities [33,34], which may lead to poor or even adverse patient outcomes [35,36].

Contingency management (CM) is a common strategy that arranges systematic application of the preferred behavior and withholds reinforcement of undesired behaviors [37]. By using points, levels, prizes or other rewards, and well-established reinforcement schedules, individuals are actively engaged, and therefore CM can encourage individuals to practice behavioral change in a novel and entertaining way facilitating increased generalization to the person’s natural environment and enhancing motivation for change. CM interventions have been effectively utilized in numerous substance use interventions to facilitate adherence to programs and increase compliance to treatment, allowing participants to develop the needed skills to improve their quality of life [38,39,40,41]. While CM strategies have been shown to be efficacious with several populations [42,43,44,45], few studies have assessed the use of these strategies in the treatment of CP [46].

As noted above, past research has indicated poor adherence in clinical trials for individuals with NF1, and the limited availability of tailored mobile applications for their disease. Therefore, the current study attempted to address the prevalent gaps in research by employing contingency management to increase adherence and engagement of a tailored mobile application for NF1. The aim of the mixed methods study was to understand the effect of contingency management on the engagement of a self-management chronic pain mobile application called iCanCope (Toronto, ON, Canada) specifically tailored for adults with NF1. This paper focuses on program engagement in the iCC and iCC + CM groups that received the intervention.

## 2. Methodology

### 2.1. Participants

Participants were recruited from NF advocacy agencies within the United States (NF Network, NF Northeast, NF Texas, and NF Central Plaines) through social media postings and direct mass emails. Participants were included in the study if they met the following inclusion criteria: (1) adults who were 18 and above with a diagnosis of NF1; (2) able to read and understand English at 5th-grade level; (3) permanently reside in the United States and (4) have pain interference aggregate scores of three or more in the last two weeks using the Brief Pain Inventory-Short Form (BPI-SF) scale. Participants were excluded if they had: (1) an undiagnosed case of NF1; (2) a documented major co-occurring psychiatric disease; (3) moderate to severe cognitive deficits using the SCID-5; or (4) depression assessed using the Patient Health Questionnaire (PHQ-9) (10 = mild major depression cut off score) or anxiety assessed using the Generalized Anxiety Disorder scale (GAD-7) greater than or equal to the appropriate thresholds; (10 = moderate severe anxiety cut off score).

### 2.2. Procedure

A three-arm randomized clinical trial (RCT) with a block randomization to control for gender was implemented. Groups were randomized into one of three conditions: Treatment as Usual (TAU), iCanCope-NF (iCC), and iCanCope-NF + CM (iCC + CM). This paper will focus on program engagement in the iCC and iCC + CM groups that received the intervention. The study was registered at clinicaltrials.org (NCT04561765).

### 2.3. iCanCope Mobile Application

iCanCope is an evidence-based mobile app (iOS/Android) designed to empower individuals with CP to better manage pain and improve function [47]. The mobile app integrates principles of CBT to facilitate empowerment and build independence in pain management through improved knowledge, self-efficacy, and coping skills [48,49], and has demonstrated effectiveness in multiple populations [47,50,51]. The iCanCope application was systematically tailored by the lead author to incorporate specific material for NF1, including an understanding of NF1, findings from published articles, content on Acceptance and Commitment Therapy (ACT), mindfulness-based stress reduction activities, and yoga videos.

#### 2.3.1. iCanCope Orientation

At study intake, individuals in the two intervention conditions (iCC-NF and iCC) received a 10-min orientation to the iCanCope program. Participants were provided with login and password information, general information about the application, instructions, and contact information for technical issues.

#### 2.3.2. iCC Group

Individuals assigned to this group had complete access to the ICC-NF mobile application for an 8-week period and were asked to check in daily along with engaging in the different components of the mobile application. As seen in Figure 1, there are five main components of the iCC-NF mobile application (Check-Ins, Pain Areas, Trends, Goals, and Library). A full description of the content available can be seen in Buono et al. [52].

#### 2.3.3. iCC + CM Group

Participants randomized to this condition had complete access to the outlined above iCanCope mobile application. However, participants were incentivized and rewarded for the following: (1) frequency of system use per day (1 point/check-in up to 10 min, double points beyond 10 min); (2) consecutive days of completing assessments (5 points for the first, increasing by 2 points for each day up to a ceiling of 15 points); (3) completion of learning modules and activities (15 points for previously unassessed modules; 5 points for previously accessed modules); and (4) consecutive days of completing goals (5 points for the first, increasing by 2 points for each day up to a ceiling of 15 points). Points were accrued through access to new sections, daily check-ins, and engagement with the mobile application. The total amount of money that could be earned by the patient over the course of the two months was 50 dollars US.

After program completion, both groups participated in a semi-structured qualitative interview that captured participant’s general thoughts on the mobile application by reviewing the different features (Library, Check Ins, Trends, and Goals), what needed to be added or improved, and likes and dislikes. All interviews lasted between 15 to 30 min and were conducted online through Zoom, with the audio file being stored for transcription.

### 2.4. Data Analysis

Quantitative data were analyzed using SPSS v23.0 (IBM). All quantitative data were extracted from the local server. Descriptive statistics were used to summarize the background characteristics of the sample and program engagement data (Abstract). The Abstract analytics platform (formerly APEEE) has been and continues to be used to analyze engagement data generated in the iCanCope application [53]. The powerful platform allows study investigators to query data generated by users in real-time and to identify mechanisms to motivate engagement. Specifically for the current study, the platform produced a visualization of check-ins, goals achieved, goals created, and articles read over time, and provided the quantitative data used for the findings presented herein. For the app, the following engagement interactions were captured: articles read, goals created, goals set, and check-ins completed. For the quantitative analysis, we conducted a one-way ANOVA to evaluate the differences between groups, where significance was set at *p* = 0.05.

Following a grounded theory approach, qualitative data were coded using a standard, qualitative procedures to code the data [54]. Two coders (FB and KL), working independently, read discharge interviews from the participants, and identified phenomena in the text that were deemed responsive to the question and thus, in the opinion of the coder, should be regarded as relevant data for inclusion in the analysis. All phrases or statements conveying meaningful ideas, events, objects, and actions were collected. If both coders selected the same phrase or statement in the answer to a given question, then it was counted as an agreement. Overall, percent agreement between coders averaged 95% and disagreements were resolved through discussion and consensus.

## 3. Results

### 3.1. Demographics

Forty-two adults were randomized in the iCC + CM group of which 6 individuals dropped out of the study and 40 adults were randomized in the iCC group of which 2 dropped out of the study. Overall adherence rates for both groups were above the a priori set percentages, respectively 86% for iCC + CM and 95% for iCC. Mean age was _M_ = 40.5, _SD_ = 13.6 for the iCC + CM group, and _M_ = 41.9, _SD_ = 12.9 for the iCC group, where both groups 66% (n = 24) for iCC + CM and 68% (n = 26) for iCC identified as a female. The remaining demographic data are presented in Table 1.

### 3.2. Quantitative

The usage across the four areas of the mobile application (check-in, articles read, goals created, and goals achieved) between groups (iCC and iCC + CM) was evaluated using a one-way ANOVA and is summarized in Table 2. There was significance for goals achieved iCC + CM (M = 29.1, SD = 63.9) compared to that of iCC group (M = 3.3, SD = 9.7), F = (1, 71) = 5.92, *p* = 0.017. The iCC + CM group (M = 11.86, SD = 18.7) was significantly more involved in goals created than the iCC group (M = 2.97, SD = 6.5), F = (1, 71) = 7.37, *p* = 0.008. The total articles read in the iCC + CM group (M = 80.1, SD = 88.2) was significantly more than the iCC group (M = 25.8, SD = 45.3), F = (1, 71), 10.91, *p* = 0.002. The total number of check-ins was not significant between the groups, F= (1, 71) = 0.786, *p* = 0.378. For both the goals achieved and created, the participant had a choice between defining their own goals (user-generated) or picking app-recommended goals. Within the goals created, user-generated goals (M = 6.3, SD = 0.9) was highly significant to that of the app recommended (M = 76, SD = 1.1), F (1, 71) = 14.21, *p* < 0.001. For goals achieved, user generated goals (M = 14.5, SD = 13.3) was again highly significant to user-generated goals (M = 1.6, SD = 2.1), F (1, 71) = 15.8, *p* < 0.001.

The nine most and nine least engaged articles across both groups are noted in Table 3. Within Table 3, the most preferred articles (n = 5) discuss alternative strategies for dealing with chronic pain, followed by (n = 3) learning about NF1 and the symptoms associated with NF (n = 3), and balancing activities (n = 1). Similarly, the least preferred articles focused on friendships (n = 4) and healthcare discussion (n = 3). Of note, each participant in both groups read the ‘treatment of NF1 pain’, article and most of all participants (71%) attempted at least one alternative strategy for dealing with chronic pain.

### 3.3. Qualitative

Of the total 74 individuals who were randomized into two groups (iCC + CM and iCC), 53 (74%) participated in interviews. Overall, 95% of participants in both groups indicated they would like to continue accessing the mobile application, and 83% enjoyed the general appearance. Four central themes were identified from qualitative interviews that included: (1) Goal feature, (2) Trend feature, (3) Articles, and (4) Feedback. Of note, CM was not indicated as a major or sub-theme in the analysis. Rather, participants in the iCC + CM group were providing feedback on other facets of the mobile application, focusing on the impact of the mobile application. Participants stated, “In these 8 weeks, it has completely changed the way that I manage my health, it genuinely has, I look at things differently, things make sense in my head more… this application will be so helpful”, and “helpful when it came to exercising, felt motivated to reach goals, accountability to stay on track, didn’t help with sleep goals”. Additionally, after conducting the grounded theory analysis, further indication was noted that the implementation of CM was effective in adhering participants to complete the study, while maintaining engagement throughout.

Participants who highly engaged with the Goals feature (46%) appreciated that the application provided suggestions for goals and that they could create personalized goals as well. These participants reported that the push notifications sent to their devices encouraged them to complete the goals daily and enjoyed a feeling of accomplishment upon marking the goal completed in the app. Some examples of goals individuals had success with included daily exercise, medication adherence, and social engagement. Participants who underutilized this feature (43%) reported struggling to complete set goals and experienced difficulty navigating this portion of the application. Reasons for difficulty in navigating ranged from goals were not allowed to be continuous (goals had to be restarted each day) to lack of desire of completing goals. Additionally, 8% of participants suggested having the option to set reoccurring and long-term goals, 3% recommended a dialogue box to elaborate on why the goal was or was not completed, and 8% requested more incentives to complete goals (e.g., rewards).

A large majority of participants (73%) reported enjoying the trends feature as it allowed them to view their weekly data and investigate correlations among the measured variables (e.g., pain, sleep, mood, physical activity). In addition, participants utilized the trends section to draw conclusions about certain activities or variables that may have influenced an increase or decrease in their pain (e.g., on Wednesday they lifted heavy objects at work, and on Thursday their pain score was higher). Moreover, four participants noted that they had a positive experience showing this feature to their medical provider to report their experience of pain over time. Some participants (7%) also reported enjoying how the Trends and Library features were linked, sharing that receiving suggestions for articles to read based on their trends data was very useful.

Approximately one-quarter of interviewed participants (26%) found the articles helpful and easy to read. Of the interviewed participants, 47% reported engaging with meditation recordings, yoga videos, and stretching exercises as means to help manage their pain. Many participants noted that they were already familiar with some of the information provided in the articles regarding NF1, but still found themselves learning new information. Given that many participants had read through all the available articles over their two-month trial, they often expressed a desire for more articles within the library on a variety of topics including body image and NF1 (n = 1), approaches to coping with pain (e.g., ACT and CBT strategies, exercises for specific areas of pain; n = 7) and up-to-date articles about ongoing NF research (n = 3), which they believed would lead to more engagement with the application.

Feedback on what could be improved for the mobile application was minimal; however, 42 (79%) participants requested some form of community support or community feature within the mobile application. Two rationales for the need for community support among the participants were: (1) to help reduce loneliness as a result of having NF1 and (2) to share experiences among others with NF1. A female participant mentioned, “No one in my family has NF, and I do not know anyone who has this disease.” This statement was echoed by another participant, “Living in the [current state] there is not a single individual that I know who has NF”. Moreover, a male participant mentioned, “sometimes you need someone to give you a hug or nod their head in agreement”.

## 4. Discussion

The current study sought to understand the impact of the integration of contingency management on engagement with a pain self-management application among individuals with a rare disease who were dealing with chronic pain. In evaluating the effect, significant differences were noted between the CM group and the Regular (iCC) group in total usage, articles read, goals established, and goals accom plished. The qualitative feedback indicated that participants endorsed the mobile application, found it easy to use, and appropriately engaging for continued usage. It should be noted that while there were more dropouts in the iCC + CM group compared to the iCC group (6 to 2), there was not a significant difference noted between the adherence of the two groups. The incentives provided to iCC + CM group in the study may not have impacted adherence, based on the qualitative and quantitative findings.

There is a wide array of mobile applications for chronic pain, and several integrated strategies to enhance engagement. Unfortunately, many of these mobile applications have not been rigorously tested [30]. The iCanCope integrates principles of CBT to facilitate empowerment and build independence in pain management through improved knowledge, self-efficacy, and coping skills [48,49]. The iCanCope app has been empirically evaluated for multiple CP adolescent populations and young adults [47,48,50]. The evidence base supports high user satisfaction, high adherence to symptom tracking, and preliminary efficacy in improving pain-related outcomes [50,51,55]. The current study adds to the previously evaluated engagement research with the iCanCope mobile application and extends the current research on NF1 chronic pain.

Despite not demonstrating a difference between the iCC and iCC + CM groups in the total number of check-ins, the total and aggregate number of goals achieved, goals created, and articles read were significantly higher in the iCC + CM group indicating the individuals in the iCC + CM group were more engaged with the mobile application than the iCC group. Moreover, by reinforcing the engagement of the mobile application, it increased the magnitude of usage by the iCC + CM group for a relatively nominal reinforcement. The benefit of contingency management, when employed appropriately, can provide clinical advantages over standard treatment or treatment without CM, as well as providing increased exposure to therapeutic options [56]. Future research should evaluate the effectiveness of monetary versus social/community feedback reinforcement on engagement and sustained usage [57]. Another important implication of the study was the difference between self-generated goals and user-provided goals. As described above, there were significant differences between the two categories; however, participants routinely and positively commented about having the ability to establish their own goals and prioritizing these goals with what they learned in the mobile application. For example, participants mentioned, using both the suggested and personalized goals, but the personalized goals were more beneficial for their mood, and discussed the ease of using personalized goals over generated goals. In doing so, participants self-managed their own chronic pain using the mobile application, which was the intention of this mobile application. By frequently utilizing the self-generated and user-provided goals, the participants empowered themselves to establish goal-oriented lives, which is a main principle of CBT and ACT, while addressing their chronic pain.

### Limitations

The current study is not without its limitations. There was an uneven distribution of female-to-male participation. We had difficulty retaining and recruiting males who wanted to participate and fit in the inclusion criterion. One potential explanation could be, as noted in other areas of chronic pain research, that men are less willing to participate in research as they are not willing to accept that they have issues or problems [58,59]. This could be especially true when it comes to alternative treatments or therapy, as opposed to medications. Moreover, we noticed there was not much diversity, as defined by race of participants, in participants in the research study, as displayed in Table 1. Thus, these concerns potentially may limit the generalization of the data. Future studies should examine modifying the inclusion criterion to increase the likelihood of participation, by expanding access to the mobile application. Another limitation of the current study could be the incentives of contingency management. Instead of social praise, payment in the form of gift cards could have shaped the behavior of the participants. Research has demonstrated that contingency management can have both positive effects (e.g., increased response rate, adherence) and negative responses (e.g., payment of responses, lack of generalization). However, given the minimal incentive (up to 5 dollars per week) over the course of the eight weeks, this would not justify the negative responses (e.g., lack of response) to the usage of contingency management.

## 5. Conclusions

The study demonstrates the impact of contingency management on engagement with a mobile pain self-management application in adults with NF1. Together, the quantitative findings and qualitative interviews strongly support incentives to encourage individuals to participate. In doing so, the individuals were exposed to more content and were more appreciative of the strategies the mobile application provided.

## Figures and Tables

**Figure 1 cancers-15-03213-f001:**
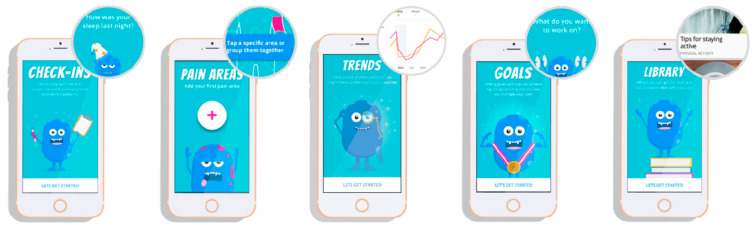
The iCanCope-NF mobile platform empowers users to track their symptoms, visualize trends, set functional goals, and learn evidence-based self-management strategies. The platform aims to help users feel empowered and independent in their self-management journey through improved knowledge, self-efficacy, and coping skills [52].

**Table 1 cancers-15-03213-t001:** Demographics study groups.

	ICC (N = 38)	ICC + CM (N = 38)	*p*-Value
Age, mean (SD)	41.9 (12.9)	40.5 (13.6)	0.663
Race (n; %)			0.558
White	33 (78%)	28 (74%)	
Black	2 (5%)	4 (11%)	
Asian	1 (3%)	2 (5%)	
Hispanic	2 (5%)	2 (5%)	
Gender (n; %)			0.443
Male	11 (30%)	12 (29%)	
Female	26 (67%)	24 (71%)	
Transgender	0	0	
Other	1 (3%)	0	
Highest Education (n; %)			0.577
Advanced Degree (e.g., Masters, PhD, MD)	4 (10%)	4 (10%)	
Bachelor’s Degree	13 (31%)	10 (26%)	
Associate degree	8 (19%)	9 (23%)	
Some of College	7 (17%)	6 (15%)	
Technical School	2 (5%)	1 (3%)	
High School	7 (17%)	10 (26%)	
Age at diagnosis in years, mean (SD)	11.7 (13.1)	10.4 (11.9)	0.664
Number of NF1 surgeries, mean (SD)	2.84 (1.5)	2.8 (1.4)	0.85
Parental History of NF1 (n; %)			0.408
yes	13 (34%)	10 (26%)	
maybe	6 (16%)	4 (11%)	
no	19 (50%)	22 (58%)	

**Table 2 cancers-15-03213-t002:** Aggregate Totals of Usage Across Groups.

ICC Engagement Category	Grouping	Total Count Across All Users	Mean per User	SD	*p*-Value
Goals Achieved	iCC	121	3.2	9.5	0.017
	app recommended	1			<0.001
	user-generated	120	3.2	9.5	
	iCC + CM	1020	29.1	63.9	
	app recommended	183	1.6	2.1	<0.001
	user-generated	854	14.5	13.3	
Goals Created	iCC	110	2.9	6.4	0.008
	app recommended	10	1.91	0.3	0.032
	user-generated	99	1.82	0.3	
	iCC + CM	415	11.8	18.7	
	app recommended	77	1.1	0.2	<0.001
	user-generated	361	6.3	0.9	
Articles Read	iCC	957	25.86	45.283	0.002
	iCC + CM	2805	80.1	88.2	
Check-ins	iCC	1996	53.9	27.1	0.378
	iCC + CM	2407	68.7	97.8	

**Table 3 cancers-15-03213-t003:** Most and Least Preferred Articles.

Most Preferred Articles	Total Count of Views	Least Preferred Articles	Total Count of Views
Treatments for NF1 Pain	250	Setting goals related to physical activity	6
What is fatigue and how can I cope with it?	191	Talking to your healthcare team	6
Reclined Twist (Yoga Pose)	174	Pacing yourself	6
Balancing School, Work and Leisure Activities	140	Maintaining a relationship with a boyfriend or girlfriend	6
Can heat or cold help my pain	136	Developing and sticking to your healthcare plan	6
Common Causes of Pain in NF1	129	3-sentence health summary	6
Relaxation and Chronic Pain	120	Dealing with setbacks	4
Rest Time	111	Tips for staying connected to family and friends	4
Progressive Muscle Relaxation with Tension	103	How to help friends understand chronic pain	3

## Data Availability

The data presented in this study are available in this article.

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
