# Peer review of "Maintaining Engagement in Adults with Neurofibromatosis Type 1 to Use the iCanCope Mobile Application (iCanCope-NF)"

_cancers, 2023, doi:10.3390/cancers15123213_

Round 1
Reviewer 1 Report (Previous Reviewer 1)
I refer to my earlier review:
"The Authors presented the results of a study registered as NCT04561765 related to the use of a mobile application iCanCope (an app that helps people with a chronic disease cope with everyday pain). In the current work the Authors evaluated the iCanCope mobile application for NF1 -patients on increasing engagement through the usage of contingency management.
The description of the course of the study was presented very carefully. I have no complaints about the methodology.
However, from the medical point of view, the work is not interesting:
- too long-winded description of the methodology,
- complicated nomenclature,
- little focus on the course of the disease
- scanty description of the study group.
I believe the use of iCanCope could be described in a medical journal as a 'Brief Report' rather than in such a long-winded article. I propose to completely modify the article - present the possibility of using the application in patients with pain, without going into details about the content of the application."
The authors did not respond to the reviewer's comments.
Reviewer 2 Report (Previous Reviewer 2)
no
This manuscript is a resubmission of an earlier submission. The following is a list of the peer review reports and author responses from that submission.
Round 1
Reviewer 1 Report
CANCERS
Manuscript ID: cancers-2286929
TITLE: Maintaining Engagement in Adults with Neurofibromatosis Type 1 to Use the iCanCope Mobile Application (iCanCope-NF)
The Authors presented the results of a study registered as NCT04561765 related to the use of a mobile application iCanCope (an app that helps people with a chronic disease cope with everyday pain). In the current work the Authors evaluated the iCanCope mobile application for NF1-patients on increasing engagement through the usage of contingency management.
The description of the course of the study was presented very carefully. I have no complaints about the methodology.
However, from the medical point of view, the work is not interesting:
- too long-winded description of the methodology,
- complicated nomenclature,
- little focus on the course of the disease
- scanty description of the study group.
I believe the use of iCanCope could be described in a medical journal as a 'Brief Report' rather than in such a long-winded article. I propose to completely modify the article - present the possibility of using the application in patients with pain, without going into details about the content of the application.
Author Response
The Authors presented the results of a study registered as NCT04561765 related to the use of a mobile application iCanCope (an app that helps people with a chronic disease cope with everyday pain). In the current work the Authors evaluated the iCanCope mobile application for NF1-patients on increasing engagement through the usage of contingency management.
The description of the course of the study was presented very carefully. I have no complaints about the methodology.
However, from the medical point of view, the work is not interesting:
- too long-winded description of the methodology,
- complicated nomenclature,
- little focus on the course of the disease
- scanty description of the study group.
I believe the use of iCanCope could be described in a medical journal as a 'Brief Report' rather than in such a long-winded article. I propose to completely modify the article - present the possibility of using the application in patients with pain, without going into details about the content of the application.
Author’s Response: We respect the reviewer’s perspective and we have shortened the methodology to reduce being long-winded. We did keep enough of the methodology to ensure for replicability and thoroughness of the design, as this study has never been conducted in the NF community before.
Reviewer 2 Report
This study investigated if contingency management (CM) can enhance the engagement of pain self-management mobile application for NF1 patients. Previous studies have shown that CM is an effective treatment for substance use and related disorders, and mobile application-based interventions for chronic pain are intensively under studying. An issue for mobile application-based pain management is engagement. Very commonly, patients lose interest in using the apps before they gain any health benefits from using them. This study tried to address if CM can enhance the engagement of the mobile app iCanCope-NF. As expected, the CM group had significant increase in using the app, such as more frequent check-ins, read more NF1-related articles, set up and achieved more goals.
1) the same research group has published an article to describe the rational and design (PMID: 35036627) of the current study but was not cited. Figure 1 is identical to figure 2 of PMID 35036627, except missing one for trends.
2) The educational level is an important factor that affects the capability and tenacity of using mobile applications. This study excluded patients with cognitive deficits, however education history should be considered and be listed in table 1.
3) whether the improved engagement of iCanCope by CM has led to a better pain self-management was not mentioned and no data has been shown in the paper.
4) In table 2, among the icc group, how many goals achieved were app recommended and how many were user generated? same question for goals created. whether the goals that were set by used are meaningful for pain management? Maybe the users (patients) just wanted to get award (points) and set up easier goals.
5)Some sentences are not easy to read. Page 2 paragraph 3 "Futhermore, they .... trained professionals": Last sentence on Page 2 paragraph 4.
Author Response
This study investigated if contingency management (CM) can enhance the engagement of pain self-management mobile application for NF1 patients. Previous studies have shown that CM is an effective treatment for substance use and related disorders, and mobile application-based interventions for chronic pain are intensively under studying. An issue for mobile application-based pain management is engagement. Very commonly, patients lose interest in using the apps before they gain any health benefits from using them. This study tried to address if CM can enhance the engagement of the mobile app iCanCope-NF. As expected, the CM group had significant increase in using the app, such as more frequent check-ins, read more NF1-related articles, set up and achieved more goals.
- the same research group has published an article to describe the rational and design (PMID: 35036627) of the current study but was not cited. Figure 1 is identical to figure 2 of PMID 35036627, except missing one for trends.
Author Response: We appreciate the reviewer’s feedback. We have modified the figure.
- The educational level is an important factor that affects the capability and tenacity of using mobile applications. This study excluded patients with cognitive deficits, however education history should be considered and be listed in table 1.
Author’s Response: We have updated Table 1 to include educational background based on the reviewer’s comments.
3) whether the improved engagement of iCanCope by CM has led to a better pain self-management was not mentioned and no data has been shown in the paper.
Author’s Response: The reviewer makes an excellent point. However, the intention of the manuscript is to highlight the importance of contingency management in facilitating treatment for NF1 populations. We are currently evaluating the iCanCope Mobile application on chronic pain.
4) In table 2, among the icc group, how many goals achieved were app recommended and how many were user generated? same question for goals created. whether the goals that were set by used are meaningful for pain management? Maybe the users (patients) just wanted to get award (points) and set up easier goals.
Author’s Response: We have updated table 2 based on the reviewer’s comment.
5)Some sentences are not easy to read. Page 2 paragraph 3 "Futhermore, they .... trained professionals": Last sentence on Page 2 paragraph 4.
Author’s Response: We agree with the reviewer, and we edited these highlighted sections, along with the remainder of the manuscript.
Reviewer 3 Report
I want to congratulate the authors on their research on this issue. This manuscript is the result of many years of study. However, I have a few concerns and suggestions to improve this manuscript.
Introduction:
I am used to the Rasopathies and understand the problems faced by patients with NF1, but I believe this is not true for the target audience. So I suggest an introductory paragraph about the disease and the causes of chronic pain. I think it will improve the clarity of the text and helps the public to reach the objective of this study.
Methods:
Figure 1: I missed an explanation about this figure as a legend.
Results:
The authors should write about the reasons for dropping out. I wondered why people in the iCC + CM group decided to quit, even with the possibilities of rewards.
Discussion:
The predominance of women kept my attention. As a disease with autosomal dominance inheritance, I expect equal distribution between sexes. So why was it difficult to find male patients? I wondered if they didn't participate in the NF advocacy agencies (Children's Tumor Foundation and NF Network) as women did. If so, why? This is a problem in other diseases and syndromes; we need the information to target it.
Author Response
want to congratulate the authors on their research on this issue. This manuscript is the result of many years of study. However, I have a few concerns and suggestions to improve this manuscript.
Introduction:
I am used to the Rasopathies and understand the problems faced by patients with NF1, but I believe this is not true for the target audience. So I suggest an introductory paragraph about the disease and the causes of chronic pain. I think it will improve the clarity of the text and helps the public to reach the objective of this study.
Author’s Response: We appreciate the feedback from the review. We have attempted to tamper the language to improve the clarity of the text.
Methods:
Figure 1: I missed an explanation about this figure as a legend.
Author’s Response: We have added an explanation of the figure.
Results:
The authors should write about the reasons for dropping out. I wondered why people in the iCC + CM group decided to quit, even with the possibilities of rewards.
Author’s Response: We were unsure as well. We were unsuccessful to reach all members of the iCC+CM group, in understanding why they did not want to participate.
Discussion:
The predominance of women kept my attention. As a disease with autosomal dominance inheritance, I expect equal distribution between sexes. So why was it difficult to find male patients? I wondered if they didn't participate in the NF advocacy agencies (Children's Tumor Foundation and NF Network) as women did. If so, why? This is a problem in other diseases and syndromes; we need the information to target it.
Author’s Response: We were generally surprised at the lack of responses of males as well; however, we were ecstatic that we had a preponderance of women at various ages consenting to the study. One potential explanation could be, as noted in other areas of chronic pain research, that men are less willing to participate in research as they are not willing to accept that they have issues or problems. This could be especially true when it comes to alternative treatments or therapy, as opposed to medications. A second potential explanation could be at the reviewer points out, more individuals who are registered on CTF are women than men. We cannot be certain of this, as we are not privy to the specific gender breakdown. We have added to the limitations about the unequal balance of women and men, along with our explanation for why this occurred.
Round 2
Reviewer 2 Report
The authors have make changes based on my comments.